# Innovative Business Process Reengineering Adoption: Framework of Big Data Sentiment, Improving Customers' Service Level Agreement

**Heru Susanto** [1,2,*], **Aida Sari** [1] **and Fang-Yie Leu** [3]

1   Center for Innovative Engineering and School of Business, Universiti Teknologi Brunei, Mukim Gadong A BE1410, Brunei
2   Center for Research Collaboration of Graph Theory and Combinatory, BRIN-ITB-UTB-UI, National Research and Innovation Agency, Jakarta 10340, Indonesia
3   Computer Science, Tunghai University, Taichung City 407224, Taiwan
*   Correspondence: heru.susanto@utb.edu.bn

**Abstract:** Social media is now regarded as the most valuable source of data for trend analysis and innovative business process reengineering preferences. Data made accessible through social media can be utilized for a variety of purposes, such as by an entrepreneur who wants to learn more about the market they intend to enter and uncover their consumers' requirements before launching their new products or services. Sentiment analysis and text mining of telecommunication businesses via social media posts and comments are the subject of this study. A proposed framework will be utilized as a guideline, and it will be tested for sentiment analysis. Lexicon-based sentiment categorization is used as a model training dataset for a supervised machine learning support vector machine. The result is very promising. The accuracy and the quantity of the true sentiments it can detect are compared. This result signifies the usefulness of text mining and sentiment analysis on social media data, while the use of machine learning classifiers for predicting sentiment orientation provides a useful tool for operations and marketing departments. The availability of large amounts of data in this digitally active society is advantageous for sectors such as the telecommunication industry. These companies can be two steps ahead with their strategy and develop a more cohesive company that can make customers happier and mitigate problems easily with the use of text mining and sentiment analysis for further adopting innovative business process reengineering for service improvements within the telecommunications industry.

**Keywords:** information retrieval; text analysis; information extraction; clustering; classification; visualization; database technology; machine learning; data-mining framework

## 1. Introduction

The way individuals communicate and obtain information has changed dramatically as a result of social media's rapid development. Social media is now pervasive, is becoming increasingly vital, and is a powerful tool for anyone, particularly in the telecommunication sector. Telecommunications firms are now using social media apps such as Instagram, Twitter, and Facebook to provide a range of services and engage with a diverse range of customers. Moreover, organizations can utilize social media to discover key internet users, assess consumer opinions about brands or special promotions, uncover prospective customer troubles or complaints, extract emotionally sensitive issues, and detect business concerns. Hence, sentiment analysis can be performed by allowing firms to modify text mining technologies for use with their social media. Sentiment analysis is the detection of positive and negative feedback in texts using natural language processing (NLP), machine learning, and data analysis; telecommunications companies can use this to monitor

customer feedback and understand customer needs. Text mining can assist the telecommunications sector in accumulating large amounts of data, analyzing them, and proposing solutions in such instances. This study reveals the conceptual framework for text mining approaches in order to conduct sentiment analysis, which converts raw free-text data into quantitative findings from social media for the telecommunications industry.

This paper organized as follows: the next section, Section 2, discusses the research problem; Section 3 is a literature review. Moreover, research methodology is provided in Section 4, followed by key findings in Section 5 and results and discussion in Section 6. Finally, the last section provides a conclusion of this study.

## 2. Research Problem

As of July 2021, the number of social media users globally was 4.48 billion, which is equal to almost 57 percent of the world's total population [1]. In addition, as social media grows in popularity among the younger generation, it becomes easier for companies to get in touch with potential customers, particularly those in younger age groups. The majority of people 16–24 years of age have used social media to research products online and discover business brands. Meanwhile, young adults in Southeast Asia [2] also utilize a diverse range of social media sites, averaging over $7^1/_2$ platforms each month, compared to a global average of 6.3 platforms for users of all ages.

Brunei's telecommunications sector serves the entire population of 439.5 thousand people by offering internet and mobile services. In January 2021 [3], there were 417.5 thousand internet users, 435 thousand social media users, and 568.2 thousand mobile phone users. Customers utilize social media to contact service providers when they experience issues such as a lost internet connection or no mobile signal and are unable to contact them through customer support. Such input will be posted on social media platforms via postings, comments, status updates, and direct messaging to share their thoughts and difficulties. Accumulating sentiments from customers via social media is important because it serves a purpose for the telecommunication company to improve their service, as the sentiments are required in order to propose answers to customers and to comprehend customer difficulties.

However, the underlying problem of this is manual scanning, which is ineffective in accommodating thousands of data points in the form of texts. It will take a significant amount of time to finish all screening. Competition is becoming more challenging—to the point that a great deal of information collected from social media, such as frequently asked questions, feedback, and opinions, as well as client identification using a survey, is not suitable. In addition, it has come to attention that the telecommunications industry has not been effectively and efficiently satisfying their customers' expectations, resulting in an outpouring of customer feedback, both positive and negative, on their social media, and it seems as if finding a solution is part of a difficult problem.

## 3. Literature Review

### 3.1. Introduction

Users of social media platforms such as Twitter, Facebook, and Instagram are known to offer their comments and feedback on a company's products and services because these platforms allow people to post their opinions about products and share their thoughts [4]. Sentiment analysis, often known as feedback mining, is a technique that employs NLP, statistics, and machine learning to extract and place feedback from a textual input into categories such as subjectivity and polarity recognition [1]. In addition, Pavaloaia, et al. described sentiment analysis in simple terms, "Sentiment Analysis is a social media tool that involves checking how many negative and positive keywords are included in text message associated with a SM (social media) post" [5] (p. 1).

The authors in [4,6] mentioned that the recognition of the requirement for sentiment analysis has been growing due to the fact there is also a rising need to estimate and structure underlying data from social media in the form of unstructured data. Text mining

is a difficult work and focuses on topical words of different topics and sentiment analysis is necessary to classify it into positive and negative polarity, and to select sentiment signals for real-time analysis [7,8]. As consumers continue to share textual material on social media, text mining and sentiment analysis are becoming more popular as mentioned by [9–11].

*3.2. State of The Art: Related Works*

3.2.1. Sentiment Analysis: Customer Dissatisfaction

Ref. [12] stated that having a data mining modeling tool can assist a business in achieving profit-making goals and advocated data mining for the sales and marketing department of telecommunication in Nigeria. They focus the sentiment analysis to gain a precise view of targeting data, the inability to translate and formulate business questions correctly and the problem of addressing data quality globally. The purpose of the research was to develop and implement the analysis that can be used to retain existing customers, attract new ones, effectively manage and allocate resources, goods, and services, and it successfully enhanced operations of the sales and marketing department.

In addition to [12], research by [13] also utilizes sentiment analysis to assess these telecommunication providers' image and reputation based on customer satisfaction obtained from Twitter data. The sentiments are identified and compared using three distinct algorithms to gauge customer satisfaction on 3G, 4G, and internet services after cleaning and data balancing. Furthermore, 5G is expected to roll out in mid-2022, raising awareness of the 5G opportunities among businesses and consumers is important [14–18]. Customers must always be informed of current changes in business processes in order to be prepared for future developments, and the best approach to determining whether customers are ready is to analyze their feedback and thoughts.

Clearly, customer satisfaction is important when doing business because the customers who are satisfied will be more loyal to the brand, product, or services the businesses provide. There are several ways that sentiments are used in research to help different businesses understand their customers and work around their products and services to continuously develop customer satisfaction. For example, the authors in [19] have concentrated on creating a priority map for tourist attractions and developing the tourism in Garut Regency using sentiment analysis, by using a method of descriptive qualitative method using text mining of Google reviews and Instagram as their platform to collect textual information. By utilizing social media sites, they managed to extract overall positive and negative sentiments and categorized their findings according to different categories, such as local infrastructure, accommodation, information. In comparison to a standard survey, text mining approaches reduced the expenses of gathering feedback and increased the firm's information discovery of feedback and opinions. They developed a user-friendly social media competitive tool called VOZIQ that can be used to gather sentiments from Twitter to analyze the different competition of two different companies and generate sentiments that are distinctive to a particular industry as standards to differentiate key performances, social media marketing efforts, and highlight the potential problem area that needs to be improved. Moreover, they have used this analytical tool to analyze the Twitter accounts of five different retail companies, and they believe it is a viable method and can be used in many business operations [20,21].

On the other hand, ref. [22] mentioned manually processing a vast amount of social web data that is rising in volume, subjectivity, and diversity has become difficult especially when technology is continuously developing. As a result, they have focused on text mining approaches in assessing the use, scope, and applicability of machine learning (ML) techniques for customer sentiment analysis in online reviews in the hospitality and tourism industries. Due to the familiarization of the digital ecosystem, telecommunication sectors are growing faster as technology use is rising, therefore reduction in workload and customer dissatisfaction is equally important to manage. There are a few examples of researchers using telecommunication industries for their sentiment analysis research. For instance, [8] performed research for the telecommunication industry using the deep learning algorithm

*Q-Meter* to detect telecommunication service complaints to monitor the difficulty of tasks and evaluate the signal strength they received from customers, which resulted in 70% of classifications of sentiment being deemed useful. The paper also proved that 92% of the complaints about the weak signals made by Twitter users were accurate and verified. Other customer complaints are also categorized for future reference in order to assist the industry in improving the quality of the service experience. Service providers managed to design strategies for customers by prioritizing services that can fit the demands of customers and, therefore, reduce customer dissatisfaction. Furthermore, it was determined to significantly help future researchers as well.

### 3.2.2. Coping Procedures: Complex Sentiments

Emotions are one of the few things that can also be considered for analyzing sentiments. Textual information in the form of emotions can be a complex procedure to detect. Therefore, a study by [23] performed opinion mining and sentiment analysis on Twitter data in order to discover four main types of textual features that have been employed in the past, including semantic, syntactic, stylistic, and Twitter-specific aspects that pointed towards the direction of future research. The analysis allows the tracking of attitudes such as irony, emotions, and quantification, which they believe have received more attention. Another emotion-based study was performed by [24], they used NLP technology to examine subjective data on customer opinions and views on social media to perform sentiment analysis. They base their research on Thayer's model, which is psychologically defined as opposed to the polarity classification used in opinion mining, and Instagram hash tags are used for their opinion gathering. In addition, the proposed method found that the accuracy rate for all sentiment categories was 90.7 percent, which indicates that the model has performed well. The proposed Twitter sentiment analysis (TSA) techniques performs in-depth analysis to aid in the discussion of related trends, the identification of intriguing unsolved difficulties, and the identification of future research opportunities.

Twitter has become an important social media platform that is widely used for text mining and sentiment analysis. Several researchers in this literature review have performed text mining through Twitter. An example report by [25] thought that using conventional methods such as surveys are expensive and time-consuming tasks to gather feedback from customers. Therefore, his research study utilizes the lexicon-based approaches to determine sentiment polarity on Twitter by combining both positive and negative words with a scoring function to analyze two giant retail stores in the UK. The methods of data access, data cleaning, data analysis and visualization were used in this methodology for text mining and sentiment analysis from R statistical software open-source tool packages. The results show their text mining algorithms may be used to assess user-submitted product and service reviews, as well as provide insight into future marketing tactics and decision-making procedures.

### 3.2.3. Analyze Failure Trends and Classify Customer Feedbacks

A methodology was proposed to three major telecommunication providers in Indonesia by [26] by using word cloud, as they believed this to be faster than the traditional sentiment analysis approaches, to capture complex relations between words while still maintaining fast summarization of trends and noise around social media. They have collected conversations about the major telecommunication providers from Twitter and each of the providers are constructed and analyzed. The purpose of this is the need of marketing intelligence for decision making about the market and its competition. Other than Twitter, a researcher from [27] used the Facebook pages of different Jordanian telecommunication brands to analyze sentiments that are in a Jordanian dialect in customer posts. All the sentiments that are gathered and analyzed are classified manually using four main classifiers: support vector machine (SVM), K-nearest neighbor, naive Bayes, and decision tree. This paper managed to show from the results that SVM outperforms the other three sentiment classifiers. The authors in [27] wanted to be able to detect whether posts published by

users were positive or negative but could only classify the comments posted rather than the posts itself.

Ref. [28] also utilizes Twitter as a platform for sentiment analysis by analyzing tweets from various companies in Saudi Arabia that were written in English. Moreover, these researchers manage to classify the sentiments using K-nearest neighbor and naive Bayes that were found trending in the days and months into positive, negative, and neutral classes. In addition, machine learning approaches such as the K-nearest neighbor algorithm were employed in this work for sentiment analysis. However, the sentiments did not include from the Arabic language which could actually provide a larger data sample for the result.

Ref. [29] provided the findings of an investigation to evaluate the effectiveness of social media posts as an indicator to underpin successful social media self-marketing tactics, which included sentiment analysis of Facebook postings. Furthermore, [29] discovered that using sentiment analysis facilitates follower negativity when user-generated activity is low, but sentiment analysis provides more consistency and complement results than simply analyzing comments, likes and shares of posts. Sentiment analysis based on machine learning was shown to be capable of assessing hundreds of thousands of comments that accompanied a single uploaded post.

### 3.3. Summary and Gap Identification

Previous studies focused on understanding and modeling specific sentiment analysis and text mining for social media, such as for enhancing organization sales and operations, marketing for competitive analysis, analyzing different companies, tourist attractions and comparing the different results to other global companies. However, there is still an absence of studies that explicitly assess the uncovering of sentiment analysis through social media such as Instagram text mining for telecommunication sectors, despite the few studies in which the majority use Twitter and Facebook.

There is evidence in previous research proving the sentiment analysis is a potential solution that can help in workload reduction and customer dissatisfaction. However, it is also limited to certain business sectors rather than telecommunication sectors. As a result of this opportunity, this paper will prioritize the use of sentiment analysis for telecommunication sectors as a way to cope with different feedback and opinions that are laid out in their social media.

In addition, the population are heavily dependent on the telecommunication sectors here for both day-to-day communication and as an internet service provider and have used Instagram as an active social media platform, as compared to other social media platforms that have been used for research previously, such as Twitter and Facebook. The comprehensive and structured strategy for dealing with consumer sentiments on Instagram, as well as efficiently identifying customer sentiments to assess failure trends in the sector, is required to be introduced and looked into so that it can be specialized for the telecommunication sector.

The language used in majority of the sentiment analysis is English, and other types of languages that are not related but have a possibility of appearing in tweets and comments have been omitted. The previous research into text mining and sentiment procedures may not work for Malay languages because of its differentiation. Instagram comments on telecommunication companies have a mixture of both Malay and English languages that can be used to determine more of the sentiment effectively with this paper's proposed framework, text mining, pre-processing, and analysis.

By studying and understanding different research on sentiment analysis, the plan to attend the gap in the research is to create a proposed framework of text mining and sentiment analysis based on Instagram that is most suitable and easy to manage and that can also reduce workload of telecommunication companies, maintain customer satisfaction, analyze trends, develop coping procedures and swiftly classify customer sentiments that caters for a mixture of both Malay and English languages. The proposed framework will be tested out for results to see how effective it is in mining texts, processing

texts, and analyzing sentiments through Instagram that are specialized for the use of the telecommunication companies.

*3.4. Theoretical Discussion*

For the theory part of the research, this paper will explain the benefits of sentiment analysis for further understanding the interests of text mining and sentiment analysis. Figure 1 represents the benefits of sentiment analysis for further understanding of this research. This chart can help readers to understand the importance of sentiment analysis and its outcomes and how this analysis can support the infrastructure, product, or service of the telecommunication sector.

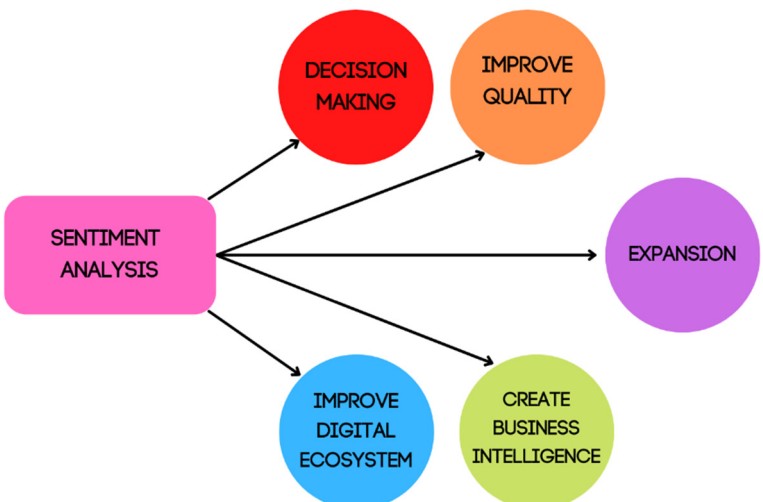

**Figure 1.** Benefits of sentiment analysis as business process reengineering tools.

- Decision Making

Decision-making is a part of a day-to-day business activity and sentiment analysis is proven to be beneficial in assisting decisions, especially in terms of marketing. Telecommunication companies distribute a lot of products and services out to their customers, one example of which is their mobile application. These products and services can actually be classified according to their positive and negative features. Customers may find that some of their products do not provide a smooth user interface and features and these companies may not know. When sentiments such as these have been analyzed, decisions can be made to help resolve such issues. Current decisions will aid in predicting the future state of the sector and revenues, therefore, spending time analyzing sentiments will reduce the unnecessary resources and time spent making unsuccessful decisions. When companies understand the benefits of sentiment analysis, they can make decisions by finding the strength of their company and use any opportunity to improve and further develop the quality of their products and services for customers.

- Improve Quality of Experience

With the use of sentiment analysis, textual features and information on social media can help detect the positive, negative, and neutral sentiment polarity of different texts. Feedback from customers is important to identify which areas in the sector require improvements and developments, so customers are always satisfied. The texts that have been gathered and their sentiments classified will help the telecommunication sector to improve the quality of experience for customers as innovative business process reengineering stages. Customers will gain trust in the companies when it comes to helping satisfy them in terms of experience or products or even customer support.

In addition, when there is less competition, companies are less committed to understanding and satisfying customers. There are just three telecommunication companies that

accommodate customer needs, and they are unlikely to be extremely competitive because all of the companies rely on one major infrastructure. With the application of sentiment analysis, however, it provides a window of opportunity to be more competitive in achieving long-term customer satisfaction and customer retention. Sentiment analysis can help in gaining better understanding of customers' emotions and feedback, especially through social media, and, most importantly, improve the quality of the experience for customers.

- Create Business Intelligence

The telecommunication sector should already be familiar with the vast and growing number of technology and social media platforms today. People are more interested in obtaining faster internet or greater availability of cheaper international calls and variety of products and services that are worth purchasing from them. Basic reporting and generating data are part of the business intelligence that is common in every business. There is potential to build a competitive advantage over their competitors by bringing information and analytical awareness of markets, competitors, consumers, operating procedures, and company performance, which are some of the many benefits of business intelligence for the telecommunication sector. By applying sentiment analysis, companies will have more awareness of the different opinions and feedback from customers and businesses, and they can use this information to build marketing strategies, provide benchmarks for their activities, forecast the demands of customers and continuously analyze the current trends circulating about relevant companies.

- Improve Digital Ecosystem

The telecommunication sectors should also be prepared as customers and users are evolving towards a more digital ecosystem. Data analytics can provide insights into customer habits and patterns, as well as better knowledge for customers, if they are made public [30–32].

This sector is also responsible for transitioning and integrating services from requiring the presence of human-to-human interaction to a more human-to-machine interaction via user interface. Taking the COVID-19 pandemic as an example, customers heavily depend on the data and internet from these telecommunication companies. Therefore, with the use of sentiment analysis, this sector can focus on tracking APIs and web scrapping through social media and websites and find strategies by understanding customer and user sentiments and emotions to build a digital ecosystem that can easily be accommodated to such situations [33–36].

- Expansion

With the presence of text mining and sentiment analysis, telecommunications can look further into the market to analyze the possibilities that require expansion in the future. It is critical to continuously move forward with the fast growth of technology instead of staying in the present and focusing solely on the current stage of the telecommunication sector. Sentiment analysis can also aid in forecasting the future for this sector by analyzing the trends that are widely talked about in the community. This allows the sector to think ahead into the future and be ready with solutions of multiple problems that may arise.

Expansion is not limited to internal growth, but the telecommunication sectors can also externally expand through methods such as merging, joint ventures and even partnerships with other local industries or globally. This will allow telecommunications companies to distinguish themselves, come up with new and improved offers, and develop value-added business models for customers. Customers will likely be happier and more satisfied knowing their service providers are doing something that can help improve their state in the industry.

## 4. Research Methodology

The primary research approach for this study is a literature analysis and the development of a conceptual framework for text mining that includes information extraction,

classification, visualization, lexicon-based sentiment classification and machine learning sentiment classification. The primary data for this research will be crawling and extracting the data from the Instagram profile pages of telecommunication sectors for further analysis. This article will then examine comments which are written in both English and Malay languages.

A proposed framework will be illustrated as a guideline for the process of this project framework for text mining and sentiment analysis for the telecommunication sector. Figure 2 shows the proposed framework of this project which consists of text crawling, extraction, and cleaning from Instagram accounts of telecommunication companies, pre-processing stage, sentiment analysis stage and data evaluation stage.

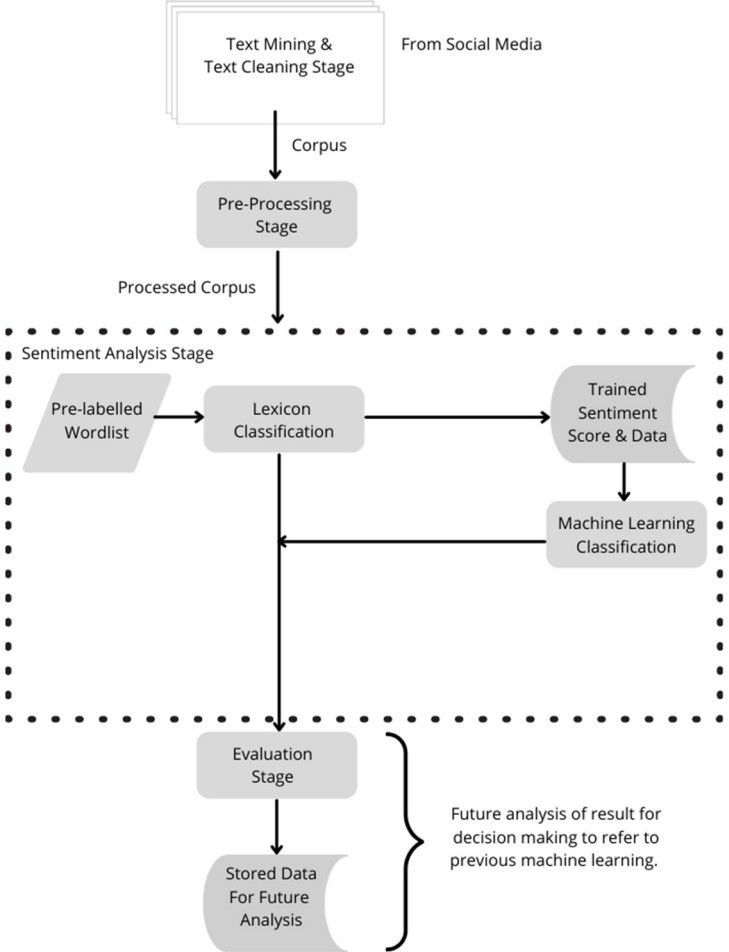

**Figure 2.** Proposed Framework.

Text mining and cleaning stage is the first stage for sentiment analysis, it is a part of the process in order for the sentiment analysis to work. Texts or comments will be extracted from the social media of the telecommunication companies, which in this case is Instagram. Once the text has been extracted, it will go through a cleaning process which will be further discussed. The corpus will be going through a pre-processing stage before it enters the sentiment analysis stage, this means the texts are prepared in a standard format, for example, all letters are in lower case, and no numerical symbols. Major analytical software such as R, RapidMiner, and Phyton language were used here for further analysis.

At the sentiment analysis stage, this research will be looking at different types of sentiment classification analysis modeling approaches, as well as their characteristics in terms of how helpful it will be to classify the sentiments accordingly. A pre-labeled wordlist will be prepared for the lexicon-based classification and this classification will be evaluated

and be used as training data and as the score for machine learning classification. Once the machine is trained, it will show the level of accuracy of the sentiment classification such as negative and positive comments.

The results will be evaluated in the evaluation stage for comparison, analysis of failure and success trends, forecasting the future trends and demands of customers and users, and improving the service, product, and quality of experiences in the industry. The stored data of every sentiment analysis will be used as the reference in the future to help develop a better understanding of the current situation of the industry and providing more structured solutions for customers. Furthermore, new data extracted from social media can refer to previous machine learning for sentiment analysis after the text mining and pre-processing stages.

### 4.1. Dataset

This study accumulates all data using various modules such as posts and comments from social media of Instagram. Table 1 depicts the number of posts and comments that have been extracted for this research. The findings show a total of 8947 comments have been collected through Instagram, with Company A having the majority of comments with 3301 (47%). Almost similarly, 3267 (47%) belong to Company B, whereas 805 (12%) belong to Company D. Lastly, there is Company C for which only 1574 (23%) of the comments have been extracted. The numbers of comments extracted have not been cleaned and pre-processed for analysis.

**Table 1.** Summary of extracted comments.

| Company | Posts | Comments |
| --- | --- | --- |
| A | 65 | 3301 |
| B | 65 | 3267 |
| C | 52 | 805 |
| D | 65 | 1574 |

### 4.2. Data Labelling

The 'category' attribute will be used to determine which category each occurrence in the dataset belongs to; these categories include sentiments and inquiries.

However, the process of text mining is not always straightforward, and sentiment analysis is one of the methods that aids in the effort of screening statements for positive or negative polarity. The 'sentiments' category contains terms that express the customer's opinions towards the telecommunication company's service provider. The 'enquiry' category comprises various queries that users have written on telecommunication firms' Instagram posts in order to receive a response.

This research paper will be looking into sentiments which will be categorized into 'positive', 'negative' and 'neutral' according to the wordlists. A sentiment that is positive indicates that customers are happy and receive satisfaction from the service or product while negative sentiments indicate the contrary. For instance, considering internet or mobile data, 'slow' is a negative word and 'great' is a positive word. The result of whether a sentence is positive, negative, or neutral will depend on the sentiment scores of the words that appear in the text, the greater the sentiment score, the more positive the sentence will be. It is crucial to keep in mind that each comment from posts has significance in defining which group of sentiment it will be classified in. There are words and phrases in comments that can help indicate the negative or positive sentiments.

Comments with the sign '?' will be classified as an enquiry and will be placed in the 'enquiry' category. This can make it easier for telecommunication firms to distinguish between opinions and questions. This function will also help detect the questions that have been extracted from the thousands of data from Instagram. However, the main focus of this paper will be the sentiment category, therefore, the results and discussion section will omit the enquiries, but they can be compiled for future reference to create frequently asked

questions (FAQs). The categorizing of the words that are considered negative, positive, and neutral in this paper will be performed manually including the English and Malay languages that frequently occur in comments.

### 4.3. Proposed Framework Stages

- Text Mining and Cleaning Stage

Within this research, the majority of telecommunication providers have already incorporated social media platforms and applications and are using social media to keep customers up to date on what is happening on their end. This stage will focus on data collection from social media using the number one platform as well as platforms that are utilized by telecommunication firms, hence, Instagram will be used for this part of the data analysis.

Figure 3 shows the data collection and cleaning stage. The extraction of comments and posts are performed manually and extraction only occurs according to a certain number of comments from the posts. When texts are extracted and crawled, the texts are expected to be noisy and in random sentence and symbols format, therefore cleaning the data is required by using manual selection to remove posts that retrieve empty comments, duplicated comments, and to ensure that only posts in English and Malay languages are included (Figure 4).

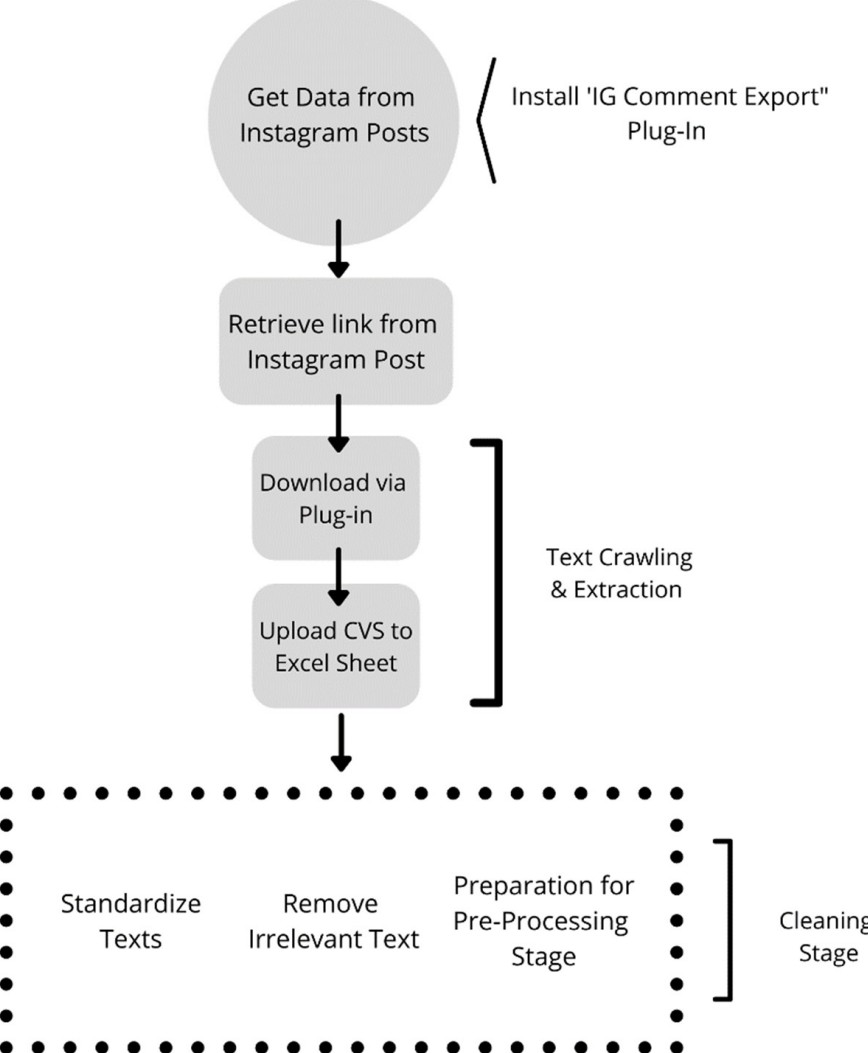

**Figure 3.** Data collection workflow.

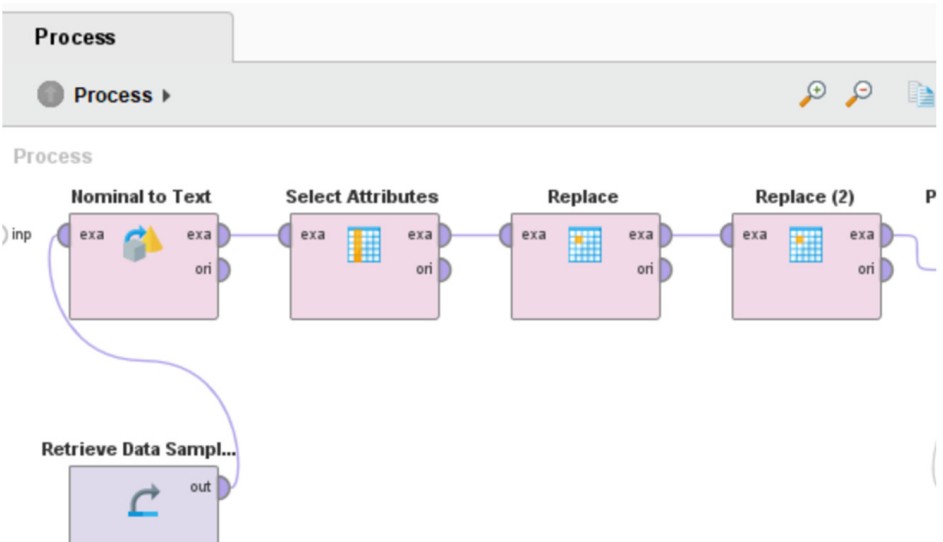

**Figure 4.** Data cleaning.

- Pre-processing Stage

    Figure 5 shows the pre-processing methods. To transform all the comments into lower case is to use the function '= lower(text)', the 'text' value can be achieved by highlighting the comments column for simultaneous transformation to lowercase letters. Tokenization of the comments is achieved by highlighting the comments column and selecting the 'Data' tab, then selecting 'text to columns', then 'delimiters' and choosing 'comma' and choosing the destination of the result and clicking 'finish' to see the result. The data that have been pre-processed will then be ready for sentiment classification.

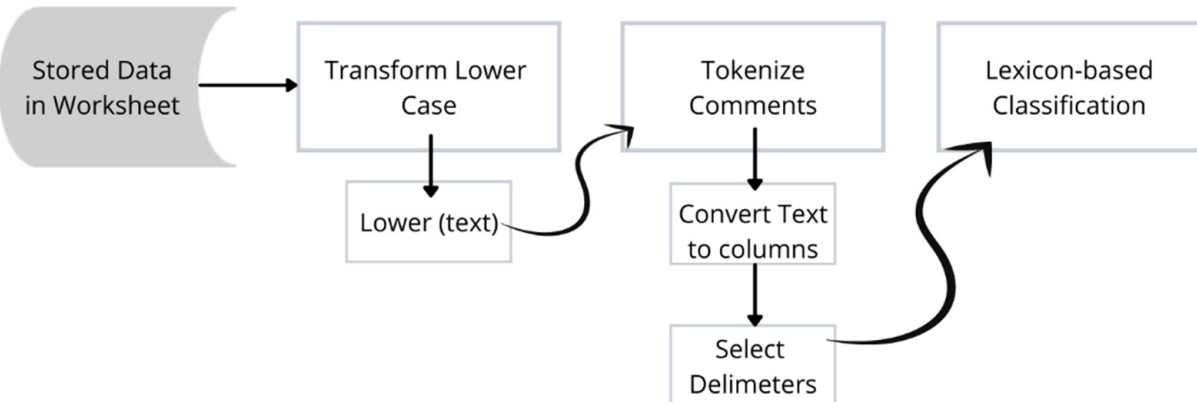

**Figure 5.** Pre-processing.

    Figure 6 shows the pre-processing stage once the data have been extracted and cleaned. Functions such as tokenize, transform cases, filter stop words in English, filter stop words in dictionary and, in this case, Malay language stop words files were used, along with filter tokens by length, as simple pre-processing operator techniques. These operator functions reused for removing all delimiters, changing letters to lowercase, eliminating numbers as well as stop words, and limiting the characters of each word to a minimum of 4 and a maximum of 20 characters in this example. This process is easier and simpler compared to manually pre-processing data. The data were also subjected to a tf–idf (term frequency–inverse document frequency) analysis. This function is used to reflect the importance of the words in the comments.

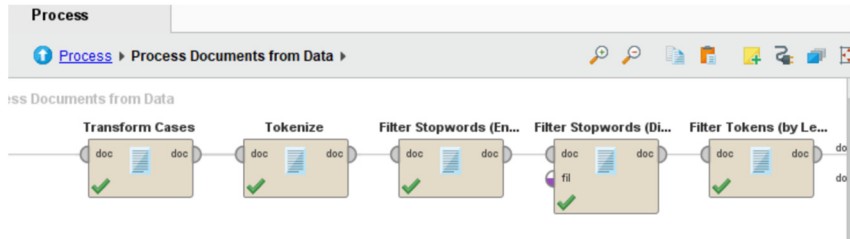

**Figure 6.** Pre-processing stage.

● Sentiment Analysis Stage

This stage of the process will be using both lexicon-based sentiment classification and machine learning-based sentiment classification. This stage is focused on measuring the polarity and classification on the body of work.

The Lexicon-based approach in this study required a huge number of words that had been pre-labeled by writers, as well as a personal constructed wordlist. The wordlists are divided into two categories: negative and positive. Table 2 shows the example of the word categorization for this research. The amount of negative and positive words on the personally made wordlist is 54 and 54, respectively. The top 50 most commonly appearing words were used to create the negative and positive terms.

**Table 2.** Wordlist of labelled words.

| Negative | Positive |
| --- | --- |
| Slow | Thank You |
| Lagging | Well done |
| Poor | Appreciate |

Figure 7 depicts the process of Lexicon-based sentiment classification, the extracted and pre-processed comments will be compared to the wordlist that was pre-labeled. This will be used for the sentiment classification to determine sentiment polarity. Negative words such as 'slow and lag' are examples of words would appear in comments as well as positive words such as thank you and appreciate. For further understanding, the Lexicon-based classifier employs sentiment score techniques to determine whether a statement is negative, positive, or neutral. The greater the number of negative words that appear in the sentence, the higher the chances it will detect negative sentiment scores on a comment. To find the sentiment score, the difference between the numbers of positively and negatively assigned words is considered. Table 3 shows an example of the score calculation.

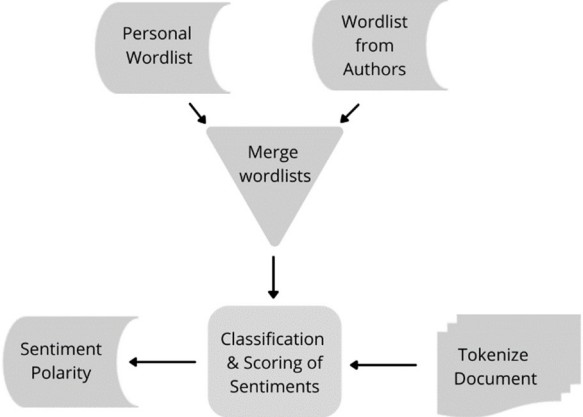

**Figure 7.** Lexicon-based workflow.

**Table 3.** Example of score calculation.

| Comment | Sentiment Classification | | | Sentiment Score and Polarity | |
|---|---|---|---|---|---|
| | Negative | Positive | No. of Words | Score | Polarity |
| The Internet is really slow and lagging!! | 2 | 0 | 7 | −0.285 | Negative |

The sentiment score can be calculated as follows:

$$\text{Score} = \frac{\text{Positive words} - \text{Negative words}}{\text{Total No.of words}} \tag{1}$$

The polarity scores for each comment are as follows:
Sentence is considered positive when sentiment score > 0,
Sentence is considered negative when sentiment score < 0,
Sentence is considered neutral when sentiment score = 0

Here, this study used another method for classifying the sentiment of comments, which is to utilize machine learning using a trained dataset that has already been tested and labeled with sentiment classes. For model training, machine learning will be reliant on manually labeled words. The support vector machine (SVM) classifier is used in this study to train the machine with labeled data from past documents in order to show how the dataset can be used to predict the sentiments of individual comments in comparable publications in the future. Machine learning will identify the frequent occurrence of words in the comments and categorize them according to their sentiment polarity.

Figure 8 shows the word cloud of the most occurring and frequently used words in the comments. The words from the word cloud can also be included for sentiment analysis for machine learning-based sentiment classification. Following the application of the basic operators, the operator 'wordlist to data' is used to list the number of words in the processed documents, while the operator 'Sort' assists in determining which attribute to include and whether the total number of occurrences should be in ascending or descending order. To show only the top 20 terms that appeared often in the document, the 'Filter Example Range' is introduced to the process. Figure 9 depicts the word listing procedure.

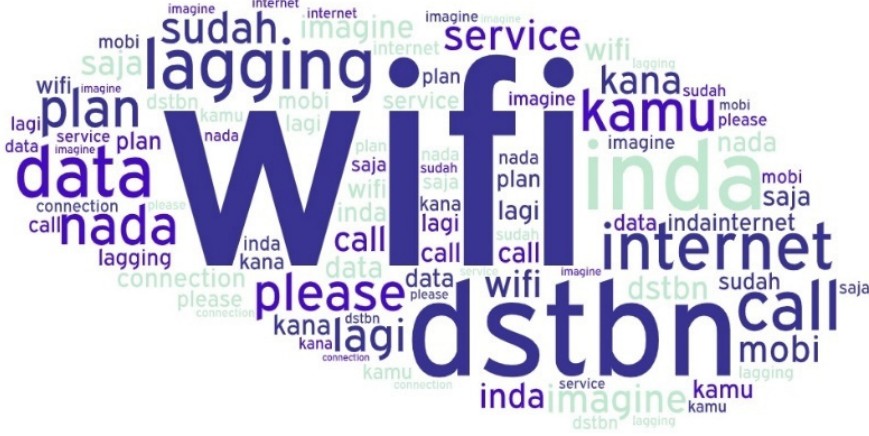

**Figure 8.** Word cloud as word list.

The word cloud can be included as a training data for classification training, however, it may not be necessary. The machine learning sentiment classification workflow is depicted in Figure 10. It is also critical to train the machine to learn on a regular basis, utilizing a variety of datasets created by the user or downloaded from the internet, to ensure that the sentiment categorization is always up to date, and that the sentiment score is accurate.

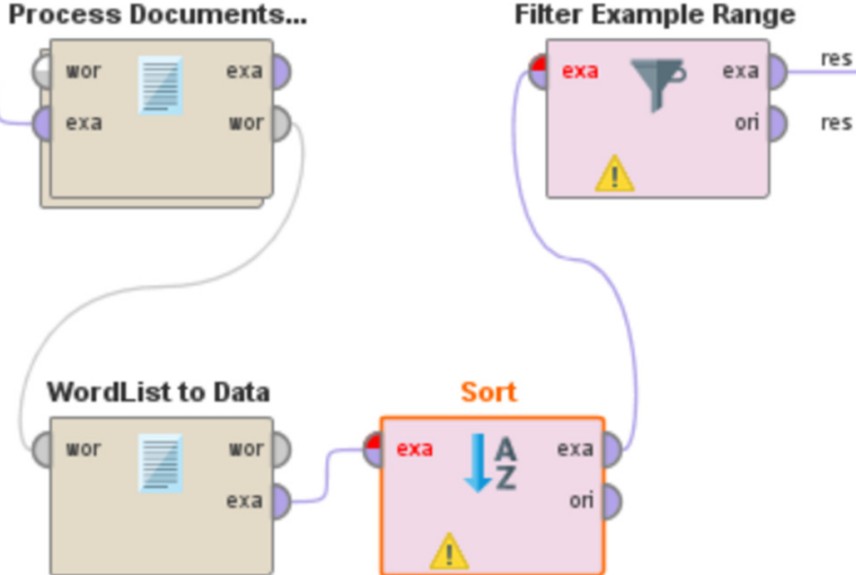

**Figure 9.** Word cloud stemming.

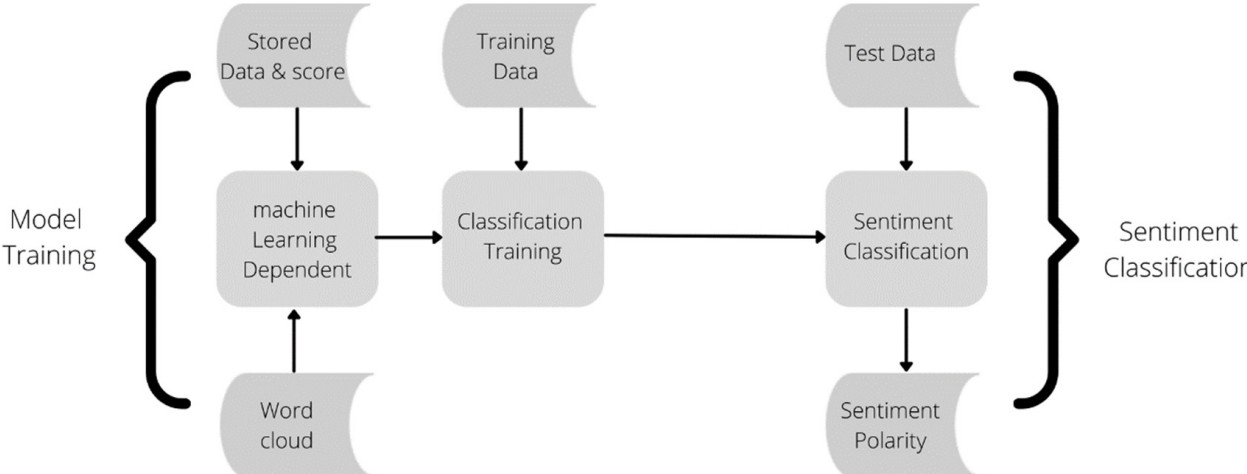

**Figure 10.** Machine learning workflow.

*4.4. Evaluation*

The classification results will be exhibited using charts such as bar graphs to depict the polarity measures, and the results will be from personal judgment. Simple distribution tables will be used to describe the classifier output. A comparison between lexicon-based sentiment classifier and machine learning sentiment classifier will also be included to view the effectiveness of different classification methods.

## 5. Results and Discussion

*5.1. Results*

User comments and views were collected from the Instagram platforms of four telecommunication firms to show the utility of the suggested framework. The company names are changed to letters to maintain their privacy. The posts and comments extracted do not include any posts that consist of giveaways and product advertisements. Table 4 shows an overview of the comments that have been extracted and cleaned. There are a total of 65 Instagram posts that have been retrieved for comments from telecommunication businesses, with a total of 7026 comments after data cleaning.

**Table 4.** Overview of obtained comments.

| Company | Total Number of Extracted Posts and Comments | |
|---|---|---|
| | Posts | Comments |
| A | 65 | 2717 |
| B | 65 | 2298 |
| C | 52 | 538 |
| D | 65 | 1473 |

The complete body of work was tokenized, filtered with stop words eliminated, and converted to lower case functions in order to extract only individual word properties. During the pre-processing step, a weighing scheme based on $tf - idf$ was applied. The pre-assembled vocabulary is then integrated with Malay and English language social media slang, as well as author-supplied wordlists, a word cloud, and a pre-labeled wordlist.

$$Tf - idf = \frac{\text{Number of times word appears in a document}}{\text{Number of documents have the words}} \tag{2}$$

The four companies were subjected to a lexicon sentiment analysis. Figure 11 depicts the outcome. The graphs for the four companies demonstrate the polarity scores of negative, neutral, and positive scores, with ratios of 1:1:0.45, 1:1.8:1, 1:4:1.75, and 1:1.7:1, respectively.

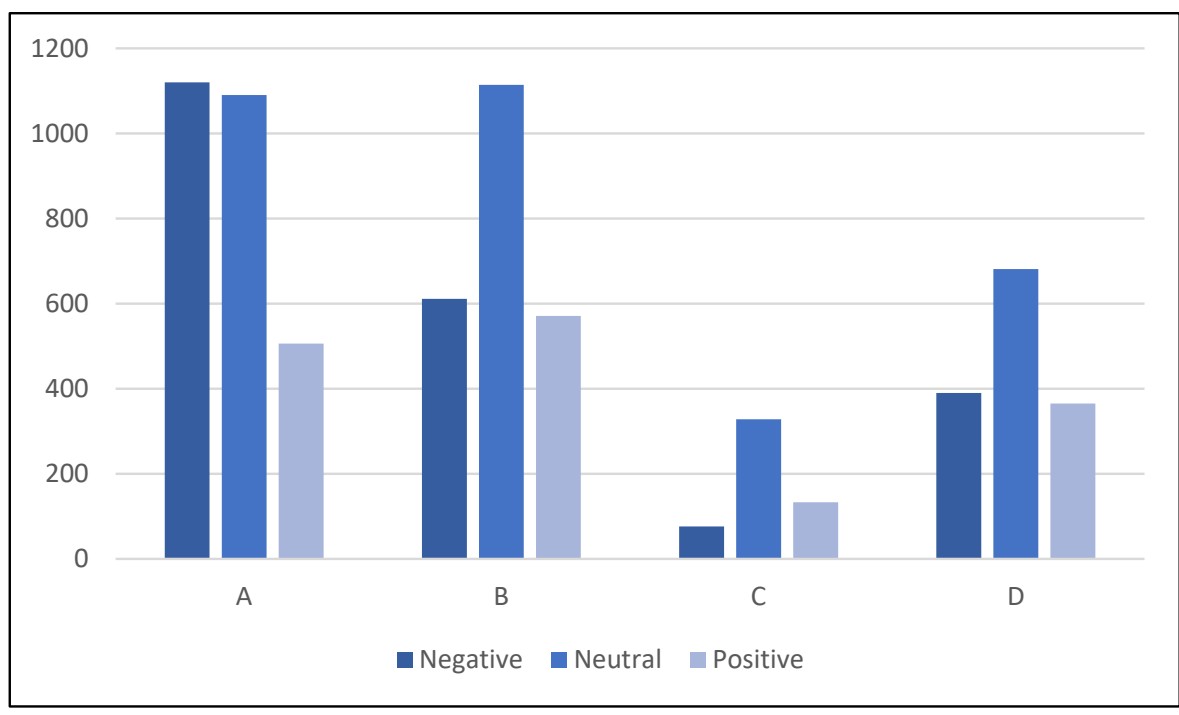

**Figure 11.** Comparison of scores for Company A, B, C and D.

Using the SVM classifier for machine learning, the results of the Lexicon-based approach are compared to the machine learning approach. The goal of this comparison is to show how successful the proposed structure and activity of sentiment analysis are. Since SVM does not allow polynomial sentiment classes, which in this case is 'neutral', this comparison will only use negative and positive sentiment results (Table 5). The total lexicon-based sentiment classification result for positive and negative is 3772. This result is split into training set 80% and testing set 20%. Table 6 compares and displays the results of the classification. As shown in the table, the negative sentiment scores are extremely close to each other, while the positive sentiment scores have a difference of 58 points. There are

potential factors that may lead to the difference between the two classification methods for positive scores. Figure 12 shows the comparison of polarity scores on a bar graph.

**Table 5.** Sentiment polarity of four companies.

| | Sentiment Polarity | | |
|---|---|---|---|
| **Company** | **Negative** | **Neutral** | **Positive** |
| A | 1120 | 1090 | 506 |
| B | 611 | 1114 | 571 |
| C | 76 | 328 | 133 |
| D | 390 | 681 | 365 |
| Column Total | 2197 | 3213 | 1575 |

**Table 6.** Sentiment polarity scores between lexicon and machine learning classification.

| | Sentiment Polarity | | |
|---|---|---|---|
| **Methods** | **Negative** | **Neutral** | **Positive** |
| Lexicon | 497 | 257 | 754 |
| SVM | 439 | 315 | 754 |

SVM: Support vector machine.

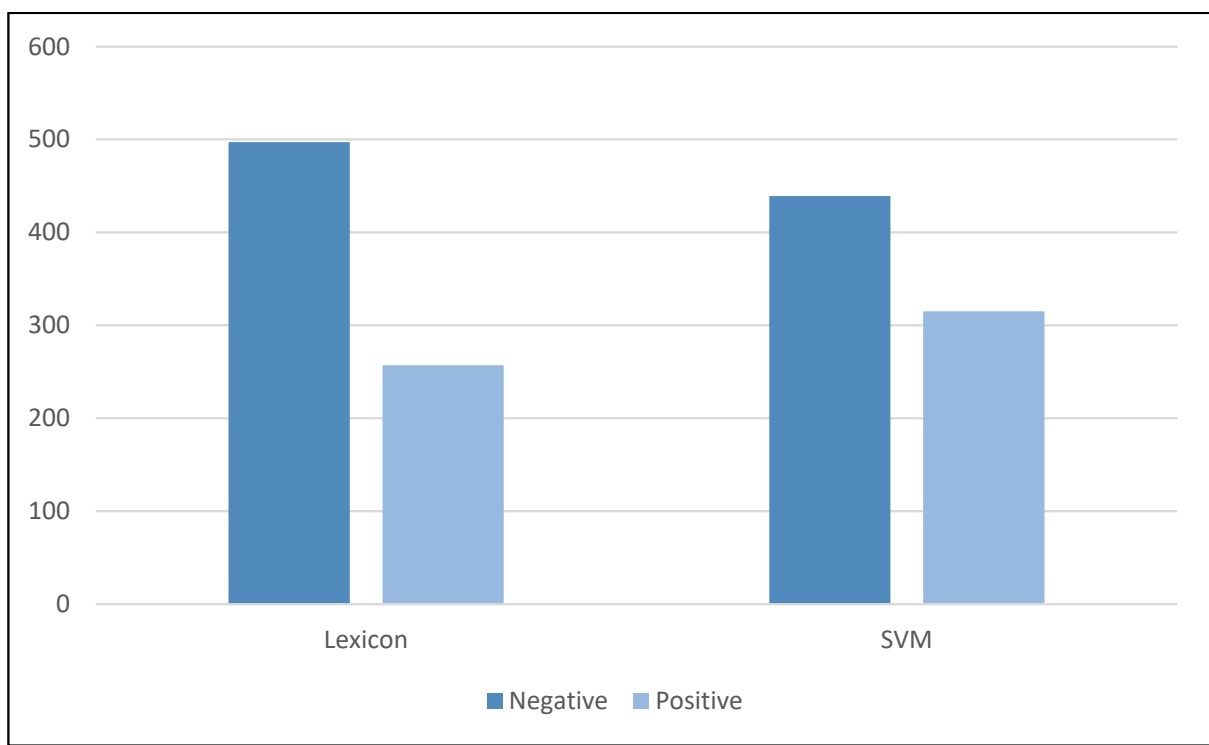

**Figure 12.** Comparison of polarity score chart.

This study has performed lexicon-based sentiment classification on 6985 comments of telecommunication companies. The sentiment polarity findings of random comments are shown in Table 7. To make things simple, scores greater than 0 are considered positive and less than 0 are considered negative.

**Table 7.** Random preview of polarity results.

| | Sentiment Classification | | | Sentiment Score and Polarity | |
|---|---|---|---|---|---|
| **Comments** | **Negative** | **Positive** | **No. of Words** | **Score** | **Polarity** |
| WIFI is so lagging!!! | 1 | 0 | 5 | −2 | Negative |
| Come on, you can do better. | 0 | 0 | 6 | 0 | Neutral |
| Thank you for dropping the price and give more data, you the best. | 0 | 1 | 12 | 1 | Positive |

To test of accuracy of performance using the support vector machine (SVM), only positive and negative sentiments were used which came to3772 in total. Within that number, the 3772 comments were split into 80% (3018 comments) for training set and 20% (750 comments) for test set so that the classifier can be evaluated on its accuracy classification. The results are shown in Table 8. The total accuracy for the classification is 59.81%. This can be used as a benchmark for future classifications with datasets of less than 1000 results being tested.

**Table 8.** SVM classifier result one.

| | Actual Results | | Row Total |
|---|---|---|---|
| **Predicted Results** | **Positive** | **Negative** | |
| Positive | 67 | 55 | 122 |
| Negative | 248 | 384 | 632 |
| Column Percentage | 21.27% | 87.47% | |

From the table, it shows that 384 out of 439 (87.47%) comments were correctly predicted as negative and were actually classified as negative, while 248 out of 315 (21.27%) comments were incorrectly classified as negative and were actually classified as positive. This is due to the small-number sample set that was used for accuracy testing which consisted of 750 comments out of the 3772. In this case, this paper has decided to test the accuracy using double the number of data as it was necessary to prove that the more data that is used for testing, the higher the accuracy rate. As such, 1500 data were then tested for sentiment classification accuracy using SVM. Table 7 shows SVM classifier result two.

Looking at Table 9, it proved a higher classification accuracy of 79.39%. Only 79 out of 879 (9%) of the comments were incorrectly classified as positive but turned out to be negative, while 232 out of 630 comments (36.8%) of the comments were incorrectly classified as negative but turned out to be positive. This can be used as a benchmark for future classification for data with more than 1000results being tested.

**Table 9.** SVM classifier result two.

| | Actual Results | | Row Percentage |
|---|---|---|---|
| **Predicted Results** | **Positive** | **Negative** | |
| Positive | 398 | 79 | 83.44% |
| Negative | 232 | 800 | 77.52% |
| Column Percentage | 63.17% | 91.01% | |

*5.2. Discussion*

The telecommunication sector has advanced significantly throughout the years with different offerings for customers and businesses as well. Their significant changes have also brought various opinions from different people. Therefore, there is a need to perform text mining and sentiment analysis. The results of the sentiment analysis proved that the telecommunication sector has plenty of improvements to make. The negative words being portrayed in comments represented the lack of quality in customers' experiences using

the internet and mobile data of these companies. When telecommunication companies understand this problem, they will be able to easily establish contingency plans and coping procedures to quickly mitigate the situation. Sentiment analysis has also helped in classifying customer feedbacks swiftly and precisely. Moreover, using the word cloud helps to show the frequently occurring words in overall texts that have been extracted as a visualization of what customers find important about the topic. In addition, the accuracy difference has also proved that more data will significantly increase the percentage of accuracy the sentiment classifier is. Therefore, in order for this framework to be highly successful, more data need to be tested for better results. The majority of the negative words mentioned about the quality of the data customers are using have not been up to their expectations. Other than that, the positive words occur only once every eight negative words that appear in comments from multiple users of social media, but the positive words may indicate sarcasm instead of sincere comment.

Furthermore, the process and result of this research would not have been successful without the guideline from the proposed framework. The framework is the skeleton of a workflow process that can be used for future reference and, with the presence of the framework, it helped to create precise outcomes. If there is any confusion throughout the process, users of this framework can refer back and indicate the fault and problems. Sentiment analysis and text mining are digital innovations that provide strong data analysis perceptions, that do not require surveys or word of mouth feedback in order for telecommunication sectors or any business know what their customers are feeling about their service, experience, and product. It reduces workload for the companies to obtain feedback from their customers.

Finally, with the help of sentiment analysis, telecommunication sectors can make better decisions in preparing for future developments in their industry, such as infrastructure, merging or collaborating with international firms that are professional in dealing with internet or mobile data issues, introducing more affordable products and services.

## 6. Conclusions

In the past years, social media such as Instagram has seen a huge growth in its platform, and due to that growth, telecommunication companies can increasingly seek ways to mine information about customers or about how other people think and feel about their products and services. This paper provides results that signify the usefulness of text mining and sentiment analysis on social media data while the use of machine learning classifiers for predicting the sentiment orientation provide a useful tool for the operations department, marketing department and more. In this research paper, we emphasized the importance of text mining for sentiment analysis and how it has reduced the workload needed to gather the feedback and opinions of customers and users through comments of Instagram. Using sentiment analysis, customers who are dissatisfied with the quality, product or services of the companies are detected, from which the trends can be compiled to use for making improvements.

With the right methods, tools, and processes, the text mining methods used in this research have helped to build coping procedures for complex sentiments that can cater to telecommunication-specific companies. By providing the simple and straightforward proposed framework in this research paper as a guideline, text mining can be performed easily for sentiment analysis. Feedback and comments are then used in sentiment analysis to swiftly and precisely analyze failure trends and customer experiences with the companies.

The availability of large amounts of data in this digitally active society can be used to such advantage in sectors such as the telecommunication industry. These companies can take two steps ahead of their strategy and develop a more cohesive company that can make customers happier and mitigate problems easily with the use of text mining and sentiment analysis.

*Challenges and Limitations*

This part of the paper will describe the limitation and challenges that are faced during the process of this research. The challenges and limitation are not limited to what is mentioned, however, these are just a few that need to be addressed that may cause or affect the overall results and processes.

- Periods of Company Establishments

Different companies were established in different years, making it difficult and unfair to see some of the different opinions that customers have towards the specific companies. Newly established companies were heavily criticized for their performance and users that are intolerant to the performance and quality of the companies have taken the comments as a way to express their judgements. Due to that, some posts received an unnecessary number of spammed comments that were not true opinions which needed cleaning.

- Comment Filter on Social Medias

There are possibilities that some posts from their social media had their comments filtered, only keeping attractive ones as a way to cover their need to respond and answer customer questions. As a result, each post has a small number of comments, and extracting 1000 comments from postings older than three years takes a long time.

- Languages

There are different languages used and the majority are Malay and English languages that are easily detectable. However, because some languages cannot be classified as Malay or English, forecasting is difficult. The comments will have to be removed because they are irrelevant to the objectives of this study.

**Author Contributions:** Conceptualization and Methodology, H.S.; Software and Validation, A.S.; Investigation and Formal Analysis, F.-Y.L. All authors have read and agreed to the published version of the manuscript.

**Funding:** This research was funded by the Universiti Teknologi Brunei (UTB) Internal Grant. Under Center for Innovative Engineering and Society Enterprise Research Thrust UTB, grant number: UTB Internal Grant 10.

**Institutional Review Board Statement:** Not applicable.

**Informed Consent Statement:** Not applicable.

**Data Availability Statement:** Not applicable.

**Acknowledgments:** Heru Susanto as main contributor and lead author. The remaining as contributors, Aida Sari, and Fang-Yie Leu. All authors read, reviewed, and approved the manuscript.

**Conflicts of Interest:** The authors declare no conflict of interest.

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
