# Peer review of "Innovative Business Process Reengineering Adoption: Framework of Big Data Sentiment, Improving Customers’ Service Level Agreement"

_2504-2289, doi:10.3390/bdcc6040151_

Round 1

Reviewer 1 Report (Previous Reviewer 1)

1. The introduced changes significantly improved the quality of the article.

2. I have no objections to the current content of the article.

3. congratulations.

Author Response

Dear Reviewer,

Thank you for your constructive feedback and comments. 
We have improved the paper and carefully proof read it.

We highly appreciate your comments and definitely they lead us to significantly improve the paper. 

Furthermore, we also accommodate to restructuring the paper as your advice and mentioned. The most significant amendments are formal and content argument to help us interpret the Introduction, Literature review, Result, discussion, Conclusion and Recommendation, Parts, as suggested. 

The most significant amendments are formal and content argument to help us 
interpret the findings. We have improved the paper and carefully proof read it. 
We think with reviewers suggestion the paper looks much better.

Honestly, we learn from reviewer comments to improve our paper.

Cordially,

Dr. Heru Susanto - Author

Reviewer 2 Report (New Reviewer)

The paper entitled 'Adopting Innovative Business Process Reengineering: Telco Big Data Sentiment, Improving Customer’s Service Level Agreement' proposes a framework to make a sentiment analysis and text mining of telecommunication businesses via social media posts and comments. In my opinion, the work is not very innovative and has many imperfections that must be eliminated:

1. I think the title should be changed and be more focused on the outcome of the work, i.e. the framework.

2. The work definitely needs to undergo linguistic correction. Among the imperfections found, the following can be distinguished:

a) On page 2 there is a different citation style.

b) Abbreviations were introduced several times (eg NLP). The full name and abbreviation should be given once, and then only the abbreviation should be used.

c) In many places, spaces are missing, sometimes there are unnecessary periods and commas.

d) Incorrect citation was used in some places - e.g. in the 'as mentioned by (He, Wu, Yan, Akula, & Shen, 2015).' I think the parenthesis is unnecessary.

e) You wrote 'et all.' instead of 'et al.'.

f) I think that instead of 'Naïve' there should be 'Naive' and ' K-Nearest Neighbors' instead of 'K Nearest Neighbor,'.

g) You wrote 'Table 5. shows the pre-processing methods'. Didn't you mean the figure? Is there a missing reference in the text to Table 5?

3. At the end of the introduction, there is no description of the article structure.

4. I believe that section 3.4 should be included earlier and should include references to literature (after all, it is in the Literature Review section). Additionally, I believe the sentence 'This chart can help readers to understand the importance of sentiment analysis and its outcomes and how this analysis can support infrastructure, product, or service of the telecommunication sector' is unnecessary.

5. The reference to equation (2) in text is missing. Since you use the tf-idf abbreviation in the text, then equation (2) should describe tf-idf and not Tf-idf. In subsection 5.3 you wrote 'TF-IDF'.

6. There is no reference to literature in the discussion.

7. Looking at the figures, it seems to me that you used RapidMiner. Am I right? If so, you should include this information in the text to facilitate the reproduction of the proposed framework.

Author Response

Manuscript ID            : BDCC-1844761

 Title    : Innovative Business Process Reengineering Adoption: Framework of Big Data Sentiment, Improving Customer’s Service Level Agreement

Cover letter

::.. Thank you for your constructive feedback and reviewer comments.

 We have improved the paper and carefully proof read it. We highly appreciate your comments and definitely they lead us to significantly improve the paper.

 Furthermore, we also accommodate to restructuring the paper as your advice and mentioned. The most significant amendments are formal and content argument to help us interpret the Literature review, Result, discussion, Conclusion and Recommendation, Parts.

 The most significant amendments are formal and content argument to help us interpret the findings. We have improved the paper and carefully proof read it.

 We think with reviewers suggestion the paper looks much better. Honestly, we learn from reviewer comments to improve our paper.

Round 2

Reviewer 2 Report (New Reviewer)

Some of my comments were taken into account, however:

1. There is a lack of space before '(' in 'population(Dadzie et al., 2014)'.

2. The abbreviation tf-idf has been introduced twice (see pages 12 and 15), and it has three spellings - 'tf-idf', 'Tf-idf' and 'TF-IDF'.

3. You should remove the dot after 7 'Table 7. shows SVM classifier result two'.

4. I still don't see any reference to literature in the discussion.

5. I also believe that the contents of the Key Findings section could be moved to Research Methodology.

Author Response

Manuscript ID            : BDCC-1844761

 Title    : Innovative Business Process Reengineering Adoption: Framework of Big Data Sentiment, Improving Customer’s Service Level Agreement

Cover letter

::.. Thank you for your constructive feedback and reviewer comments.

 We have improved the paper and carefully proof read it. We highly appreciate your comments and definitely they lead us to significantly improve the paper.

 Furthermore, we also accommodate to restructuring the paper as your advice and mentioned. The most significant amendments are formal and content argument to help us interpret the Literature review, Result, discussion, Conclusion and Recommendation, Parts.

 The most significant amendments are formal and content argument to help us interpret the findings. We have improved the paper and carefully proof read it.

 We think with reviewers suggestion the paper looks much better. Honestly, we learn from reviewer comments to improve our paper.

This manuscript is a resubmission of an earlier submission. The following is a list of the peer review reports and author responses from that submission.

Round 1

Reviewer 1 Report

Comments on the structure and form of the article:

1)      The article is not following the MDPI format;

2)      There are significant blank spaces (half of page) in the article that need to be eliminated;

3)      The content and format of tables and figures description is inconsistent with each other and the form. There are different fonts in different sizes and formats. In addition, various errors were found in the numbering of tables and figures (two tab. 3, tab. 2 before tab. 3);

4)      The titles of chapters and subsections are not consistent with each other, but also with the MDPI format. There are different styles of creating subsections, errors in numbering, frequent use of different fonts, etc;

5)      Paragraph indentation sometimes occurs and sometimes not;

6)      References in the text do not match;

7)      The equations are not numbered;

8)      English revision required.

 The overall structure and form of the article is highly incorrect and should be corrected before publication. There are too many errors to list them all here.

Comments on the substantive part of the article

The research presented by the authors is not new (not counting the use of social-media data from telecommunications companies). The methodology presented by the authors as well as their tools are outdated. In my opinion, the results could be better using the latest approaches, and the dataset could also be expanded to include more than one platform (authors used only data from Instagram). The authors also spent a significant part of the article on a detailed description of often unimportant issues.

In my opinion, the article needs revision.

Reviewer 2 Report

The paper should deal with a very interesting topic in which many researchers have specialized in the last decade. However, the paper lacks several considerations:

1) Nowadays, those who deal with text analysis need to know the new state of the art by integrating the models already present in the hugging face library. This is completely missing here.

2) Using Excel and RapidMiner does not provide the opportunity to extend the framework to a Big Data approach. It makes no sense to use such old-fashioned tools when other, more complete libraries are available in Python or R.

3) Preprocessing was fine 20 years ago. Now there are more complex libraries for stemming and lemmatization of tokens. Also, the RapidMiner images are completely unusable, even more so without showing the different settings. The code in a journal must be available via GitHub.

4) It looks like a student paper for a basic programming course essay. I do not consider it worthy of a journal publication.

5) The use of SVM without explanation of the parameters, the division into training and tests without consideration of the imbalance of the classes, the performances without comments and given as integers without proportions to the total.

6) The visualization of the results should be done with heatmap, graphs, umap, tsne etc.

7) In the literature, there are a variety of local and global weights to weigh the importance of words in the collection of documents. Here they are not mentioned.

8) I recommend that the authors review the state of the art using previously tested models and try to improve them through transfer learning. This work could be presented in a workshop for non-bigdata and non-experimental text applications.

Minor comments:

- The tables need to be placed on one page, not between two pages, which makes reading unnecessarily difficult;

- the images must have a nice texture, so they are not so grainy;

- using Rapidminer images is useless, especially without settings and with the block error view (probably just to take a screenshot).

Reviewer 3 Report

I recommend a revision based on the below points.  Please, add a point to point response to each comment in your revision:

-          I am not convinced about the novelty of the manuscript. 

-          The abstract is not technical and needs to clearly highlight the research gap.

-          The abstract also missed statistical information about the results.

-          The structure of the paper is vague. The paper needs to be restructured.

-          Don't add heading over heading. Add a few lines related to the detail of a particular section before starting a sub-section.

-          Proofread your paper from a native English speaker. There are many typos and grammar mistakes. 

-          At the end of the Introduction section, add the contributions clearly. See this paper for reference and citation ' A Novel Framework for Prognostic Factors Identification of Malignant Mesothelioma through Association Rule Mining.

-          Related work/background/literature review should have a threat to a validity section. At the start of the background section, add a threat to a validity section. In that section, state the search strings and databases that have been explored to find the related work. See the below papers for references and citations 'Performance comparison and current challenges of using machine learning techniques in cybersecurity' and 'A Survey on Machine Learning Techniques for Cyber Security in the Last Decade'.

-          The literature needs to be sub-divided into multiple sub-sections.

-          Add the below papers to your literature: A Model to Enhance Governance Issues through Opinion Extraction, An efficient stop word elimination algorithm for Urdu language, Comment Extraction using Declarative CrowdSourcing(CoEx Deco), A Novel Approach to Data Extraction on Hyperlinked Webpages

-          Add the statistical test (t-test/p-test/ANOVA, whichever is applicable) to compare your method with others. 

-          Comparison with the state-of-the-art is missed. You need to compare your method with the ground truth. 

-          You need to add a separate section for limitations of your approach and Future directions. 

-          It needs to add the reasons why these metrics are used for comparison. 

-          Add the discussion related to the time complexity factor of AI models. See this paper for reference and citation 'Cyber Threat Detection Using Machine Learning Techniques: A Performance Evaluation Perspective'.

Overall, the paper has many inconsistencies, and the contributions are not clear. The results are not compared with the ground truth properly. Limitations are not provided in their current approach. Future directions are not clearly stated.

I am looking forward to seeing your revised version. 

All the best.